# The Effect of Interaction NGF/p75^NTR^ in Sperm Cells: A Rabbit Model

**DOI:** 10.3390/cells11061035

**Published:** 2022-03-18

**Authors:** Cesare Castellini, Simona Mattioli, Elisa Cotozzolo, Alessandra Pistilli, Mario Rende, Desirée Bartolini, Gabriele Di Sante, Laura Menchetti, Alessandro Dal Bosco, Anna Maria Stabile

**Affiliations:** 1Department of Agricultural, Environmental and Food Science, University of Perugia, Borgo XX Giugno 74, 06100 Perugia, Italy; cesare.castellini@unipg.it (C.C.); simona.mattioli@unipg.it (S.M.); elisa.cotozzolo@libero.it (E.C.); alessandro.dalbosco@unipg.it (A.D.B.); 2Department of Medicine, Section of Human, Clinical and Forensic Anatomy, School of Medicine, University of Perugia, P.le Lucio Severi, 1, Sant’Andrea delle Fratte, 06132 Perugia, Italy; mario.rende@unipg.it (M.R.); desirex85@hotmail.it (D.B.); gabriele.disante@unipg.it (G.D.S.); anna.stabile@unipg.it (A.M.S.); 3Department of Agricultural and Agri-Food Sciences and Technologies, University of Bologna, Viale Fanin 46, 40138 Bologna, Italy; laura.menchetti7@gmail.com

**Keywords:** rabbit semen, NGF, p75^NTR^, apoptosis, viable cells, sperm parameters

## Abstract

Background: Nerve Growth Factor (NGF) plays an important role in the reproductive system through its receptor’s interaction (p75^NTR^). This paper aims to analyze the impact of NGF p75^NTR^ in epididymal and ejaculated rabbit semen during in vitro sperm storage. Methods: Semen samples from 10 adult rabbit bucks were collected four times (n = 40) and analyzed. NGF was quantified in seminal plasma, and the basal expression of p75^NTR^ in sperm was established (time 0). Moreover, we evaluated p75^NTR^, the apoptotic rates, and the main sperm parameters, at times 2–4 and 6 h with or without the administration of exogenous NGF. Results: Based on the level of p75^NTR^, we defined the threshold value (25.6%), and sperm were divided into High (H) and Normal (N). During sperm storage, p75^NTR^ of H samples significantly modulated some relevant sperm parameters. Specifically, comparing H samples with N ones, we observed a reduction in motility and non-capacitated cell number, together with an increased percentage of dead and apoptotic cells. Notably, the N group showed a reduction in dead and apoptotic cells after NGF treatment. Conversely, the NGF administration on H sperm did not change either the percentage of dead cells or the apoptotic rate. Conclusion: The concentration of p75^NTR^ on ejaculated sperm modulates many semen outcomes (motility, apoptosis, viability) through NGF interaction affecting the senescence of sperm.

## 1. Introduction

In addition to the already well-known role of Nerve Growth Factor (NGF) as a neurotrophin involved in the regulation of neuronal survival and differentiation [1], recent studies showed a ubiquitous distribution of NGF in different districts, including the reproductive system. The suggested biological significance of this uncommon localization is also shared among several animal species. In particular, in bovines, NGF exerts a luteotropic effect [2], and in llamas, NGF was identified as an ovulation-inducing factor protein [3,4]. NGF modulates endocrinal events, controlling reproduction in both induced [5] and spontaneous [6] ovulatory species. In particular, in the reproductive system of male rabbits, we demonstrated that NGF is expressed by different cell types: Leydig, Sertoli, germinal cells, and prostate cells [7].

NGF may trigger several processes generally mediated by the interaction with two receptors: tropomyosin receptor kinase A of 140-kDa (TrKA) and the NGF receptor of 75-kDa (p75^NTR^) [8], with high and low affinity, respectively. TrKA transduces the classical pathways, commonly downstreaming growth factor receptors: mitogen-activated protein kinase (MEK/MAPK), extracellular signal-regulated kinase (Erk), phosphatidylinositol 3-kinase (PI3K), and phospholipase C gamma [9]. The p75^NTR^ is a cell death receptor, a member of the tumor necrosis factor receptor superfamily [8], which is able to induce either (i) apoptosis, through c-Jun N-terminal kinases/caspases 3, 6, and 9, or (ii) survival, via nuclear factor kappa-light-chain-enhancer of activated B cells (NF-kB) [10], both depending on the binding with TrKA. Previous studies described the role of NGF and its receptors on sperm properties, focusing on its dose- [11,12] and time-dependent effects on sperm survival, apoptosis, and motility [13]. As shown also for other non-neurological compartments, the biological functions of NGF in sperm are mainly related to the interactive receptors involved, i.e., TrKA is maintained nearly stable during sperm storage, and modulates viability and sperm acrosomal reaction [14], whereas p75^NTR^ strongly increases throughout an 8 h storage, which seems correlated to its apoptotic role, as demonstrated in other cell lines, e.g., brain [15,16,17].

The modulating role of NGF on reproductive activity has been shown in several animal model studies and humans [6,18,19,20]. Specifically, in rabbit bucks, we previously demonstrated that where NGF and its receptors have been described in epididymal and ejaculated sperm, p75^NTR^ is mainly in the midpiece and tail, whereas TrKA resides in the head and acrosome [13]. In addition, the concentration and functions of seminal NGF showed some discrepancies among different animal species [7,21,22]. For instance, the age of rabbit bucks, the collection rhythm, and other not-yet-defined factors deeply affected the levels of NGF and its receptors in the semen [23].

This paper aims at deepening our understanding of some factors (i.e., receptors proportion, endogen NGF concentration, exogenous NGF addition) involved in the role of NGF in the rabbit reproductive system, analyzing the concentration of NGF, and focusing on the impact of p75^NTR^ in epididymal and ejaculated rabbit semen. This work may contribute to the understanding of the role and biological effects of NGF and p75^NTR^ in issues of fertility and reproduction.

## 2. Materials and Methods

If not otherwise specified, all chemicals were purchased from Sigma Aldrich (St. Louis, MO, USA).

### 2.1. Experimental Design

Different experiments were performed (Figure 1):

Exp 1. Quantification of NGF concentration and p75^NTR^ expression in epididymal and in ejaculated sperm, and the effect of p75^NTR^ expression at time 0 on the main sperm outcomes.Exp 2. Effects of exogenous NGF (100 ng/mL) during storage (up to 6 h) on ejaculated sperm.

### 2.2. Animals and Semen Sampling 

Ten healthy New Zealand white rabbit bucks aged from 10 to 24 months were raised in the experimental facility of the Department of Agriculture, Food and Environmental Science of Perugia (Italy) and used for semen collection. Four consecutive semen samples were collected (two per week) for a total of 40 samples (10 rabbit bucks × 4 replicates).

Animals were not subjected to stressful treatment that caused pain or suffering, and semen collection was performed weekly using a doe-like dummy and an artificial vagina maintained at 37 °C internal temperature. Specific guidelines for rabbit bucks [24] and the International Guiding Principles for Biomedical Research Involving Animals [25] were followed. Animals were bred in compliance with the 2010/63/EU Directive transposed into the 26/2014 Italian Legislative Decree. 

The collection of epididymal sperm was performed in a slaughterhouse, from three rabbit bucks of the same genetic strain, through washing with 1 mL of saline solution for each epididymal region (from caput, corpus, and cauda) and directly recovered in 1.5 mL tubes ready for FACScan analysis. 

### 2.3. Semen Handling 

Immediately after collection, the sperm concentration was measured using a Thoma–Zeiss counting chamber and a light microscope (Olympus CH2, Tokyo, Japan) with a 40× magnification. 

An aliquot of the semen (about 0.5 mL) was centrifuged at 700× *g* for 15 min to obtain seminal plasma (SP) and to quantify NGF concentration by using ELISA.

An aliquot of each semen sample derived from the different collections was diluted with a modified TALP [13] to achieve a final concentration of 10^7^ sperm/mL. The effect of storage was evaluated at different time points (0-2-4-6 h) in semen samples supplemented at time 0 with 100 ng/mL exogenous NGF (Merck, Milan, Italy), according to the dose–response curve previously described [13], and compared to vehicle samples diluted with PBS instead of NGF. The samples were evaluated for motility, viability, and necrotic/apoptotic processes as described below. Furthermore, receptor expression was also evaluated.

### 2.4. NGF Quantification in Seminal Plasma

Seminal plasma was collected from the above-described semen samples. Semen collection was performed by means of an artificial vagina that was kept at 37 °C and filled up with heated water at 39–40 °C, following centrifugation at 700× *g* for 15 min. The NGF concentrations in seminal plasma were detected by enzyme-linked immunosorbent assay (ELISA), according to the manufacturer’s instructions for the DuoSet ELISA for NGF (R&D System, Milan, Italy). The standard curve demonstrated a direct relationship between optical density and NGF concentration. All samples were run in duplicates. The NGF concentration was expressed in pg/mL (detection limit 31.25 pg/mL) [7].

### 2.5. Motility (CASA Analysis) 

Automated semen analysis (model ISAS, Valencia, Spain) was performed according to previously assessed parameters [26]. Briefly, two drops of the semen samples and three microscopic fields were analyzed, and the kinetic properties of at least 300 sperm were recorded. All kinematic parameters were verified, but only the leading indicators (motility rate expressed as % of total motile cells on the total sperm and curvilinear velocity, VCL, expressed as sperm speed [μm/sec] in the curvilinear trajectory) were reported.

### 2.6. Sperm Capacitation and Acrosomal Reaction

The chlortetracycline (CTC) fluorescence assay was performed as reported by Cocchia et al. [27]. Briefly, a solution composed of 45 μL of sperm suspension, 45 μL of CTC stock, and 1 μg/mL concentrated propidium iodide was added to 1.5 mL foil-wrapped Eppendorf tubes. The cells were fixed by adding 8 μL of 12.5% paraformaldehyde and one drop of 1, 4-diazabicyclo[2.2.2]octane dissolved in PBS to delay the fading of fluorescence. The CTC staining of the viable sperm was examined under an epifluorescence microscope (OLYMPUS—CH2 excitation filter 335–425 and 480–560 nm, for CTC and propidium iodide detection, respectively) detecting different sperm fluorescence patterns: fluorescence over the entire head (non-capacitated cells; NCP); a non-fluorescent band in the post-acrosomal region of the sperm head (capacitated cells; CP); or absent fluorescence on the sperm head (cells with an acrosomal reaction; Ar). Two drops for every sample (n = 40) were analyzed. Three hundred sperm cell/samples were counted.

### 2.7. Viable, Apoptotic, and Necrotic Sperm Analysis 

The externalization of phosphatidylserine was performed by Annexin V Apoptosis Detection Kit (K101-100BioVision, Waltham, MA, USA). The aliquots of semen samples were washed with PBS and resuspended in 500 μL of Annexin-binding buffer (about 1 × 10^5^). After the addition of 5 μL of FITC-conjugated Annexin V (AnV-FITC) and 5 μL of Propidium Iodide (PI, 50 μg/mL) the samples were incubated and then analyzed by a flow cytometer (FACScan Calibur, Becton Dickinson, Franklin Lakes NJ, USA). The gating strategy was performed as follows: FSC/SSC dot plot was obtained from each semen sample; a “flame-shaped region” (R1) was established to exclude debris, large cells, and aggregates; 10,000 live-gated events were collected for each sample; all samples were run in duplicate. The combination of both AnV and PI allowed for the discrimination of four sperm categories: AnV^−^PI^−^ viable, AnV^+^PI^−^ early apoptotic, AnV^+^PI^+^ late apoptotic, and AnV^−^PI^+^ necrotic cells. The analysis was performed with CellQuest Software (Becton Dickinson).

### 2.8. p75^NTR^ FACScan Analysis 

The p75^NTR^ receptors were evaluated, as described in our previous paper [13], in ejaculated semen immediately after collection (n = 40) and in epididymal sperm cells (n = 3). Briefly, aliquots of 10^6^/mL of sperm cells were placed in FACScan tubes and preincubated with PBS/BSA for 30 min at 4 °C. After washing procedures (three times in PBS supplemented with 0.5% BSA), sperm cells were incubated at 4 °C for 1 h in PBS/0.5% BSA containing 2 μg/10^6^ cells of anti-p75^NTR^ (MA5-13314, Thermo Fisher Scientific, Waltham, MA, USA), then washed and labelled with a secondary antibody (ab6785 FITC conjugated for p75^NTR^, Abcam, Cambridge, UK) for 30 min at 4 °C. p75^NTR^-positive cells were quantified by FACS analysis. FSC/SSC dot plot was obtained from each semen sample. A “flame-shaped region” was established to exclude debris, large cells, and aggregates. The count of p75^NTR+^ cells was executed by plotting green fluorescence (FL1)/FITC. Briefly, the gating strategy was as follows: FSC/SSC flame shape, dot plot TrKA/SS, gating TrKA^+^ cells and histogram of distribution of p75^NTR+^ staining on TrKA^+^ cells (p75^NTR+^ vs. p75^NTR−^). Ten thousand live-gated events were collected for each sample and isotype-matched antibodies were used to determine binding specificity. The results were expressed as percentages of positive cells/antibodies used for staining (% positive cells). All experiments included a negative control incubated with Normal Goat IgG Control Mouse IgG Isotype Control (# 31903 Thermo Fisher Scientific for p75^NTR^). The analysis was performed with CellQuest Software (Becton Dickinson).

### 2.9. Statistical Design

Exp 1. Diagnostic graphics, Kolmogorov–Smirnov and Levene’s tests were used for testing assumptions and outliers. Because non-normality of the data was detected for CP and motility, log transformation was used for analysis. The p75^NTR^-positive cells variable at time 0 was categorized using the median as a cutpoint to create two groups [28,29] called Normal p75^NTR^ (N = p75 ≤ 25.6%) and High p75^NTR^ (H = p75 > 25.6%). The means of the other parameters at time 0 were then compared between Normal and High p75^NTR^ groups using unpaired *t*-tests. Data were reported as means and standard deviations (SD). Associations between parameters at time 0 were further investigated using the Pearson correlation coefficient^®^, including p75^NTR^ as a continuous variable. The correlation was considered poor if r < |0.3|, medium if |0.3| ≤ r < |0.5|, and large if r ≥ |0.5| [30].

Exp 2. Changes in sperm-quality parameters during storage were analyzed by Linear Mixed Models (LMM), including sample as subject and time as repeated factors. The LMMs evaluated the effects of treatment (2 levels: control and 100 ng/mL NGF-supplemented), p75^NTR^ group (2 levels: Normal and High p75^NTR^), time (3 levels: 2, 4, and 6 h), and the interaction of the treatment with the p75^NTR^ group, while baseline values were included as a covariate. Sîdak adjustment was used for carrying out multiple comparisons.

Statistical analyses were performed with SPSS Statistics version 25 (IBM, SPSS Inc., Chicago, IL, USA). Statistical significance occurred when *p* ≤ 0.05. The repeatability of sperm parameters was estimated with the following formula: σ^2^ buck/(σ^2^ error + σ^2^ buck). Variance components were estimated with REML procedure [31].

## 3. Results

### 3.1. Exp 1. Quantification of NGF Concentration and p75^NTR^ Expression in Epididymal and Ejaculated Sperm and the Effect of p75^NTR^ Expression on Main Sperm Outcomes

While the percentage of p75^NTR^ positivity in epididymal sperm samples steadily amounted to 40%, the p75^NTR^-positive ejaculated cells at time 0 needed a categorization. To this end, we used the median as threshold to create two groups named Normal (p75^NTR^ ≤ 25.6%) and High (p75^NTR^ > 25.6%). Table 1 shows descriptive statistics of the parameters at time 0 in the Normal and High p75^NTR^ groups. No difference was found between Normal p75^NTR^ and High p75^NTR^ groups in all parameters evaluated. Furthermore, NGF plasma levels and p75^NTR^ expression showed high repeatability of about 50.27% for NGF and 41.43% for p75^NTR^ (data not shown). Finally, the correlation analysis showed a medium negative linear association between p75^NTR+^ cells and sperm concentration (r = −0.403, *p* < 0.01, Table 2). 

### 3.2. Exp 2. p75^NTR^ Expression in Ejaculated Sperm and Impact of Exogenous NGF (100 ng/mL) during Storage (up to 6 h) on H and N Sperm Samples

Samples with High p75^NTR^ (H) at T0 and during storage showed a lower % of motility (*p* < 0.001) and NCP (*p* < 0.001), while they showed a higher % of dead (*p* = 0.001) and apoptotic cells (*p* = 0.001) than normal (N) sperm samples (Table 3; Figure 2). The Normal p75^NTR^ group showed a lower percentage of both dead and apoptotic cells after NGF treatment than control (*p* < 0.05). Conversely, the NGF addition did not change either of these values in H sperm samples.

The motility rate was not associated with the level of p75^NTR^ (Figure 2A), while NGF was a relevant factor in the sperm kinematics. In particular, the treatment induced a significant reduction in motility percentage only in samples with H p75^NTR^ compared with N p75 ^NTR^ (Figure 2E). According to motility and VCL values, sperm capacitation was also consistently decreased by the NGF treatment in the H p75^NTR^ group (Figure 2F, *p* = 0.039, 4 h and *p* = 0.0846, 6 h, two-way Anova Sîdak). In the meantime, the impact of NGF treatment on the same H group revealed a larger percentage of both dead and apoptotic cells than in the N group (Figure 2G *p* = 0.0142, 4 h; *p* = 0.0809, 6 h; Figure 2H *p* = 0.0483, 4 h and *p* = 0.0872, 6 h). 

We displayed NGF treatment and p75^NTR^ levels in Figure 3. No differences were observed either in H or N sperm cells within all the parameters (Figure 3A–F). 

## 4. Discussion

The present findings indicate that about 47.5% of rabbit semen samples, immediately after ejaculation, showed a sperm population with High p75^NTR^, while about 52.5% showed Normal p75^NTR^ values.

In these two groups of samples, the endogenous NGF levels were comparable (1591.71 ± 413.07 vs. 1621.13 ± 428.86 in N and H p75^NTR^, respectively), suggesting that NGF does not directly affect sperm characteristics, but its effect is modulated by receptors (Figure 2A–D).

Furthermore, ejaculated rabbit sperm samples with H and N p75^NTR^ levels have a different response to in vitro addition of exogenous NGF during storage. TrKA and p75^NTR^ have been already identified in sperm of the golden hamster and humans [20,21] and ejaculated sperm of rabbit [14], and these two receptors strongly modulate its effect. 

In our previews paper, we demonstrated the presence of NGF and p75^NTR^ in seminal plasma and in different cell types of the male rabbit reproductive system. In particular, p75^NTR^ was identified in both the somatic and germ cells of the gonad as well as in the glandular epithelial and stromal cells of seminal vesicles and prostate glands [7]. In sperm cells, p75^NTR^ is mainly localized in the midpiece and tail [13]. 

Within the testes, NGF is synthesized by the Leydig cells, Sertoli cells, and germinal cells at different stages of development, including spermatogonia, spermatocytes, and spermatids. A strongly positive reaction for NGF was detected in the columnar secretory epithelial cells, and in the stromal cells of the rabbit prostate. The presence of NGF and its receptors in those cell types suggests that their growth and differentiation are finely regulated by this neurotrophin via a complex autocrine and paracrine mechanism [13,22]. 

Our previous results showed that after storage, TrKA level remained almost stable while low-affinity receptors suddenly increased, connected with an increase in apoptosis [14], probably due to a process of p75^NTR^ externalization. A similar trend was also registered in leiomyosarcoma cells [32], suggesting that the role of NGF is dependent on the balance between its receptors. In particular, the distinctive pathways “death/survival” generated by NGF-receptors’ interactions were determined by the concentration (ratio) of p75^NTR^ to TrKA.

The novelty of this paper regards the fact that a high level of p75^NTR^ in rabbit sperm cells modulates the role of NGF in sperm characteristics. Many studies have shown a positive effect of in vitro NGF addition on the main sperm traits in several animal species and humans [11,14]. In particular, Lin et al. demonstrated that NGF promoted human sperm motility by increasing the movement distance and the percentage of A-grade spermatozoa in a dose-dependent manner [20]. However, the NGF receptors’ concentration was not considered by the authors. 

The body of literature reports two distinct p75^NTR^ signaling pathways, one including TrKA activation—which suppresses JNK activity—and another focused only on p75^NTR^ activity via NFkB activation [33]. When TrKA is lacking, p75^NTR^ can act as an inducer of apoptosis and then cell death [34], modulated by NGF. Thus, it could be hypothesized that the interaction between NGF and receptor is a sort of selector for semen samples of poor quality, contributing to selecting (through apoptosis) optimal sperm cells. Indeed, apoptotic processes in sperm should be distinguished from somatic cells, considering their prominent role in both selections of defective sperm produced during spermatogenesis and the programed senescence of ejaculated sperm [35].

On the other hand, the high value of p75^NTR^ found in the epididymal sperm cells could be related to the need to select/re-absorb defective sperm. Although the reason is still debated (collection order, buck), as some of these cells—candidates for the elimination during epididymal transit—are ejaculated, it is reasonable to suggest that the high p75^NTR^ value determines a rapid decline (Figure 2G,H), contributing to the sperm selection occurring in female reproductive apparatus.

At the same time, we observed reproducible levels of NGF and p75^NTR^ among different bucks, indicating that H semen donors tend to produce sperm which is highly fragile and prone to apoptosis/death (senescence, early aging). Although further studies are needed to better define NGF and p75^NTR^ values in different males, our results are relevant for fertility and reproduction issues.

## 5. Conclusions

The p75^NTR^ level in ejaculated sperm could be considered a biomarker of the quality of the semen for its ability to modulate, through NGF interaction in some sperm, quality parameters and distinct features (motility, apoptosis, viability).

High p75^NTR^ concentration in the sperm samples is related to cell senescence and poor quality of sperm. Therefore, the measurement of p75^NTR^ in the semen samples could be proposed as a screening marker for fertility-selective programs, ideally not only in rodents, after defining the distinct cut-off for each species. The present findings could also be exploited in semen conservation and thus could improve assisted reproduction techniques.

## Figures and Tables

**Figure 1 cells-11-01035-f001:**
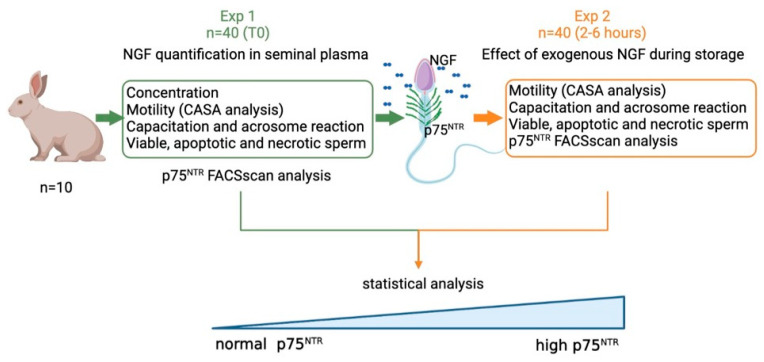
Operation map of experimental design applied on ejaculated sperm. In Experiment 1 (Exp. 1), sperm quality (concentration, motility, capacitation, viability) and receptor concentration have been evaluated in epididymal (n = 3) and ejaculated sperm samples (n = 40; 10 bucks × 4 experimental replicates). Based on the p75^NTR^-positive cells at Time 0, the samples were categorized into two groups named Normal p75^NTR^ (N = p75 ≤ 25.6%) and High p75^NTR^ (H = p75^NTR^ >25.6%). In Experiment 2 (Exp. 2), the same semen samples were divided into two aliquots: one used as control, and another added with NGF (storage times: 2–6 h), generated with Biorender.com (last accessed on 18 January 2022).

**Figure 2 cells-11-01035-f002:**
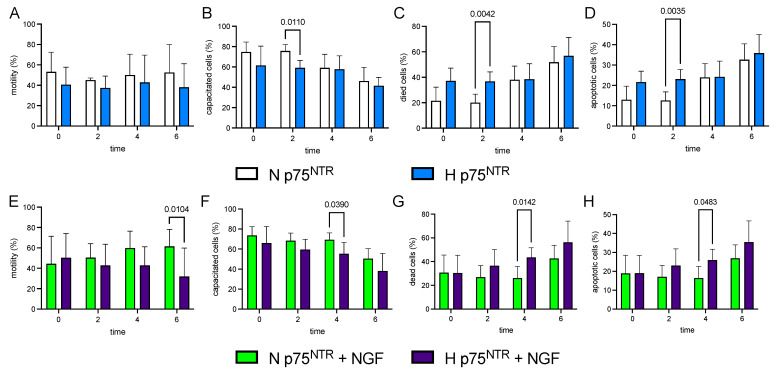
Effect of NGF treatments (100 ng/mL) on Normal and High p75^NTR^ cells during storage times (0, 2, 4, 6 h). Bar charts display Normal and High p75^NTR^-untreated (white, N and blue, H) and NGF-treated (green, N + NGF, and purple, H + NGF) for distinct time-points (0, 2, 4, 6 h) and different sperm parameters: (**A**,**E**) Motility rate (%); (**B**,**F**) Capacitated cells (CP, %); (**C**,**G**) Dead cells (%); (**D**,**H**) Apoptotic cells (%).

**Figure 3 cells-11-01035-f003:**
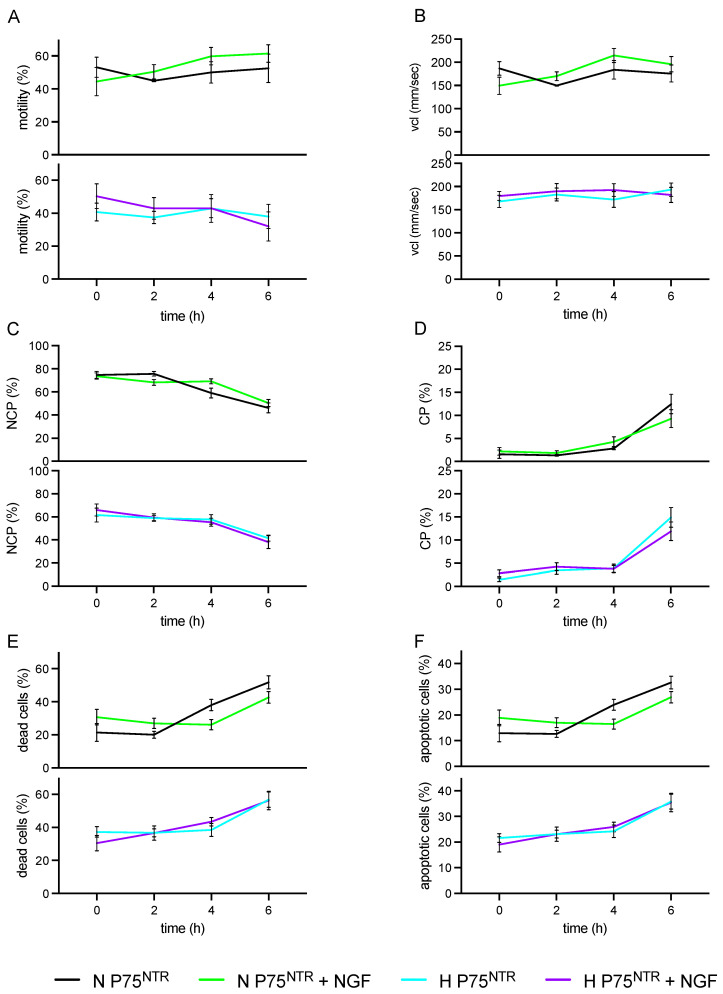
Effect of NGF treatments (100 ng/mL) on Normal and High p75^NTR^ cells during storage times (0, 2, 4, 6 h). (**A**) Motility rate (%); (**B**) Curvilinear velocity (VCL, μm/sec); (**C**) Non-capacitated cells (NCP, %); (**D**) Capacitated cells (CP, %); (**E**) Dead cells (%); (**F**) Apoptotic cells (%). Sample size n = 40.

**Table 1 cells-11-01035-t001:** Ejaculated sperm traits, NFG, and p75^NTR^ quantification in the semen of High (>25.6%) and Normal (<25.6%) p75^NTR^-positive cells at time 0.

Parameter	Group		*p* Value
	Normal p75^NTR^ (n = 21, 52.5%)	High p75^NTR^ (n = 19, 47.5%)	
**Concentration (n × 10^6^)**	234 ± 92	197 ± 87	0.198
**Dead** **cells (%)**	21.45 ± 10.76	33.88 ± 11.04	0.06
**Motility rate # (%)**	49.15 ± 23.91	42.79 ± 18.35	0.403
**VCL (μm/s)**	169.44 ± 55.71	169.12 ± 39.01	0.983
**Apoptotic cells ^†^ (%)**	15.94 ± 8.50	21.60 ± 5.38	0.06
**NCP ^†^ (%)**	74 ± 10	62 ± 19	0.092
**CP #^,†^ (%)**	2 ± 3	2 ± 2	0.388
**Ar ^†^ (%)**	0.00 ± 0.00	0.05 ± 0.22	0.331
**NGF (pg/mL)**	1591.71 ± 413.07	1621.13 ± 428.86	0.336

Values are expressed as means ± standard deviation. Abbreviations: VCL = curvilinear velocity; NCP = non-capacitated; CP = capacitated sperm; Ar = acrosomal-reacted; # analyzed after log transformation; ^†^ equal variances not assumed. Sample size n = 40.

**Table 2 cells-11-01035-t002:** Pearson correlation coefficient between parameters evaluated at the time 0.

	Dead	Motility #	VCL	Apoptotic Cells	NCP	CP#	Ar	NGF	p75^NTR^
**Concentration**	0.172	0.097	−0.140	0.261	0.011	−0.141	−0.242	−0.175	−0.403 **
**Dead**		−0.324 *	−0.233	0.951 **	−0.638 **	−0.164	0.065	−0.130	0.098
**Motility #**			0.716 **	−0.380 *	0.163	−0.258	−0.289	0.116	−0.183
**VCL**				−0.294	−0.032	−0.140	0.045	0.135	−0.071
**Apoptotic cells**					−0.583 **	−0.187	0.065	−0.170	0.062
**NCP**						−0.310	−0.203	0.008	−0.134
**CP#**							0.297	0.022	0.136
**Ar**								0.112	0.124
**NGF**									−0.119

Abbreviations: VCL = curvilinear velocity; NCP = non-capacitated; CP = capacitated sperm; Ar = acrosomal-reacted. Significant correlations with a *p* < 0.05 (*) and a *p* < 0.01 (**) are indicated with bold characters (2-tailed); # analyzed after log transformation.

**Table 3 cells-11-01035-t003:** Changes in ejaculated sperm traits, NFG, and p75^NTR^ in semen of Normal (≤25.6%, N) and High (>25.6%, H) p75^NTR^-positive cells during storage (from 2 to 6 h after ejaculation).

Parameters	NGF Treatment	Groups		*p* Value			
		Normal p75^NTR^ (n = 21, 52.5%)	High p75^NTR^ (n = 19, 47.5%)	N vs. H	C vs. NGF	N vs. H (C vs. NGF)	Time (0-2-4-6 h)
**Concentration (n. ×10^6^)**	C	222.46 ± 3.28	214.70 ± 2.87	0.156	0.870	0.236	0.125
	NGF	219.45 ± 2.95	218.69 ± 2.95				
**Dead** **cells (%)**	C	40.80 ± 2.59	42.94 ± 2.28	0.001	0.389	0.012	<0.001
	NGF	32.88 ± 2.04	46.93 ± 2.25				
**Motility # (%)**	C	46.24 ± 2.36	35.48 ± 1.64	<0.001	0.706	0.162	0.120
	NGF	56.23 ± 2.42	31.70 ± 1.50				
**VCL (mm/s)**	C	172.05 ± 10.07	180.25 ± 8.63	0.789	0.054	0.252	0.477
	NGF	200.02 ± 8.12	187.09 ± 8.82				
**Apoptotic cells ^†^ (%)**	C	25.58 ± 1.65	27.27 ± 1.45	0.001	0.279	0.028	<0.001
	NGF	20.70 ± 1.30	28.99 ± 1.44				
**NCP (%) ^†^**	C	57.91 ± 2.42	53.10 ± 2.17	<0.001	0.992	0.142	<0.001
	NGF	61.15 ± 1.94	49.90 ± 2.13				
**CP (%) #^,†^**	C	3.69 ± 0.29	4.28 ± 0.28	0.190	0.528	0.733	<0.001
	NGF	3.88 ± 0.25	5.02 ± 0.34				

Values are displayed as estimated marginal means ± standard errors. Models included baseline values as covariates. Abbreviations: VCL = curvilinear velocity; NCP = non-capacitated; CP = capacitated sperm; Ar = acrosomal-reacted; C = control; T = NGF-treated; # indicates back-transformed estimated marginal means ± standard errors; ^†^: equal variances not assumed. Sample size n = 40.

## Data Availability

Not applicable.

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
