# Peer review of "The Effect of Interaction NGF/p75NTR in Sperm Cells: A Rabbit Model"

_cells, 2022, doi:10.3390/cells11061035_

Round 1
Reviewer 1 Report
Dear authors,
please, check my comments in the attached document.
Best regards

Author Response
General Comment
Dear Editor and reviewers
Thank you for the useful comments that allow us to improve the quality of our manuscript.
In addition to the graphical abstract, we detailed the experimental plan in Figure 1 and implemented information on M&M.
Accordingly, we detailed the points requested by rev. in M&M.
In synthesis we performed two experiments:
In Exp1 we analysed 40 ejaculated samples (10 bucks x 4 experimental replicates) where sperm quality (concentration, motility, capacitation, viability) and receptor concentration have been analysed. Based on the p75NTR positive cells at Time 0 we categorized the samples into two groups named Normal p75NTR (p75≤25.6% - N) and High p75NTR (H p75>25.6% - H), using the median as cut point. Thus, the N and H p75NTR groups have been created ex-post.
Subsequently, in exp.2 we divided the semen samples into two aliquots: one used as control, and another added with NGF (0-6 hours).
Reviewer 1
The effect of interaction NGF/p75NTR in sperm cells: a rabbit model Authors describe the role of Nerve Growth Factor (NGF) and its two receptors p75NTR and TrKA in sperm physiology when storing them for a period of time. It was pointed out that there are two sperm populations in regard to the p75NTR but the NRF concentration did not vary between samples. The manuscript presents some flaws that impair the possibility of being considered for publication in a journal as Cells. In my opinion the results are not in the scope of the journal. Furthermore, the English language should be thoroughly edited and a more formal format should be used. Moreover, the methods used need to be better described. In addition, and to my mind, to consider the measurement of the levels of p75NTR as a marker of sperm quality in terms of motility, viability or acrosome integrity makes no sense when traditional evaluations are more feasible and accessible to fertility clinics. The discussion section is also very poor, the authors did not argue about the importance of the results obtained as those in seminal plasma. strongly recommend the authors reconsider the main idea of the manuscript and perhaps, look for another journal to submit the results. Please for more details, check the comments below
A: Reviewer 1 rejected the present paper; hence we followed the editor's requires, and we considered only the markers of reviewers 2-5. Anyhow, many of the Reviewer's suggestions were similar to those of the other reviewers, thus we changed the manuscript accordingly and a brief explanation of the remarks was done to Reviewer 1 in PDF files.
see also attached file

Reviewer 2 Report
Dear Editor,
Thank you for inviting me to review the manuscript.
Generally, I think that this is a well-designed interesting study and suggest to publish in your journal after the revising based on the below suggestions.
Best regards,
I would like to thank authors for their effort for this study.
My comments are;
- More information should be presented about the function, production and localization of p75 on sperm cells and production of NGF in seminal plasma (for instance which cells secrete NGF and when and how etc.)
-
I recommend to discuss the paper entitled "Nerve growth factor promotes human sperm motility in vitro by increasing the movement distance and the number of A grade spermatozoa" (doi: 10.1111/and.12375) in the text.
- Please, order the figures and tables as presented in the text.
- Supplemented figure can be presented as figure.
- In suppl fig, e and f, normal group, there is a decrease in dead and apoptotic cells at 2 hr. How do authors explain this? I can figure an increase in death cells as cells may die in the course of time, but for decrease of died cells!!!
- English should be checked to remove some errors. Please see some examples from the manuscript below.
- "During sperm storage, p75NTR of H samples showed a significant time-dependent trend." means what? increase/decrease
- "apoptosis cells" -> apoptotic cells
- "acrosomial" -> acrosomal
- "700 X g per 15 min" ->700g for 15 min ??
- Line 268, "...debated why (collection order, buck [30]) some of these..." what does that mean in brackets?
- Line 196, "... semen. in particular.." -> In particular
Reviewer 3 Report
The authors intended to study the role of p75NRT and NGF in sperm cells. Thus, it was analyzed different sperm functions in ejaculated and epidydymal sperm of rabbits.
I believe that this study has the potential to present very good results, however, the methods should be better described, so the results could be better understand.
I have very deep difficulty in understanding the group division.
I believe that the groups division should be better explained in a specific section. It is not clear the normal and high groups, how are they divided, before or after treatment? If it was after the treatment, how the cells were separated?
If it was the sample high and normal, it should also be written in the text. The rabbit that collected 4 times, was in the same group (normal or high)?
Perhaps including a diagram explaining how the samples were divided and hoe the groups were formed would be better. In addition, to name each group, so in results we know what we are looking at.
Reviewer 4 Report
This study is focused on the expression of the receptor p75NTR in sperm cells from rabbit and its interaction with NGF. However, the message is not clear, so it is difficult to evaluate the relevance of the results obtained.
All the sections of the manuscript need deep revision: 1) several sentences are difficult to understand, and 2) some terms are not correct.
Abstract section:
- Line 17. “…its receptor’s interaction (p75NTR and TrKA)”. After reading the whole manuscript, I realise that authors did not analyse the receptor TrKA, so they must eliminate it to avoid misunderstandings.
- Line 20. Please, note that “main semen traits” is not an appropriate term. Authors must refer to sperm quality parameters and specify which are these sperm parameters.
- Lines 20-21. “…in time course experiments”. What do the authors refer? Please, specify.
- Lines 18-21. The description of the Methods used must be improved
- Line 24. Please, provide the full name of NCP.
- Line 26. Please, correct “apoptotis cells” by apoptotic cells”.
- Line 30. Keywords: “semen traits”. Please, use an appropriate key word for sperm quality.
Introduction section
- Line 35. Authors indicate that NGF is present in the reproductive system. Please, indicate in which regions of the reproductive system this factor is present. Authors must support this information by citing appropriate bibliography.
- Lines 36-39. Please, revise the content of these lines and provide a clear message.
- Line 47. Please, provide the whole name of JNK.
Material and Methods section
- Line 73. Before this line authors must add the subsection: “Experimental design”.
- Lines 74-75. Exp 1.”... p75NTR expression in epididymal and in ejaculated sperm”. Did the authors locate this receptor in the plasma membrane of sperm cells? If not, authors must perform an immunocytochemical assay to determine in which region of the sperm plasma membrane locates this receptor.
- Lines 74-75. Exp 1. “…the effect of p75NTR expression on main sperm outcomes”. What do the authors refer with “main sperm outcomes”? How was this evaluated? Please, add a new subsection describing the procedure followed.
- Lines 81-82. “Four consecutive semen samples were collected (n=40)”. Considering that the study includes ten animals, I am not sure that the samples is 40. Authors must justify from an statistical point of view why they consider that the sample size is 40 instead of 10.
- Lines 89-92. Authors must provide a more detailed description of the procedure of collection of epididymal sperm. From which epididymal region the sperm samples were obtained? Please, specify.
- Lines 94-96. How many samples were analysed? And how many replicas per sample? How are the results expressed? Please, specify.
- Line 98. Which ELISA kit did the authors use to measure the NGF concentration? How many samples were analysed? And how many replicas per sample? How are the results expressed? Please, specify.
- Lines 104-105. “The samples were evaluated for motility, survival, and necrotic/apoptotic processes involving sperm cells and receptors expression as below-described.” Please, revise and improve this sentence in order to provide a clear message.
- Lines 107-108. Please, provide a more detailed description of the procedure for NGF quantification, and indicate how many samples and how many replicas per sample were analysed. How are the results expressed? Please, specify.
- Lines 113-114. What the motility rate is? Please, define. How the results of motility and VCL are expressed? How many samples and how many replicas per samples were analysed? Please, specify.
- Lines 126-127. “Three hundred sperms/sample were counted”. Please, refer to “sperm cells” instead of “sperms”. How many samples were analysed? And how many replicas per sample? How are the results expressed?
- Lines 128 and 132. Please, refer to “viable” instead of “live”. Authors must review the whole document and unify the terminology.
- Lines 129-137. Authors must indicate which sperm populations were observed and what was the staining pattern of each one. They must also indicate how many samples and how many replicas per sample were analysed. Please, also specify how the results are expressed.
- Lines 146-147. “p75NTR-positive cells were quantified by FACS analysis”. Which was the staining pattern of these cells? And the staining pattern of p75NTR negative cells?
- Lines 149-150. The results were expressed as percentages of positive cells/antibody used for staining (% positive cells). Authors must provide a more detailed information about how they calculated the percentage of positive cells in order to provide a clear message.
- Line 157. “…p75≤25.6% - N) and High p75NTR (H p75>25.6% - H). Why did the authors used this threshold? Please, specify.
- Line 163. “Changes of parameters”. Do the authors mean “changes of sperm quality parameters”? If so, please correct it.
- Line 181. “No difference was found between groups”. What do the authors refer? Please, specify which parameters were compared.
- Lines 181-185. Please, revise the content of these lines in order to provide a clear message.
- Lines 187-188. Are the results expressed as either in total numbers or in percentages? Please, specify it in the Material and Methods section. Authors must also revise this sentence in order to provide a clear message and avoid misunderstandings.
- Line 193. “kinetic traits”. I suppose that the authors refer to sperm cells. If so, it is not appropriate to refer to kinetic traits considering that the authors only included to motility parameters. Please, describe the results obtained from the analysis of sperm motility and VCL.
- Line 194. Please, refer to “were observed” instead of “were recorded” and to “sperm samples” instead of “sperm”.
- Line 196. “kinetic of semen”. Please, note that authors did not analyse the kinetic of semen by the kinematics of sperm cells, and more specifically the sperm motility and the VCL. So, you must use an appropriate term.
- Line 197 “motility”. It is not clear which motility parameter the authors measured. As indicated before, authors must indicate in the Material and Methods section if they refer to either the total percentage of motile sperm or the percentage of progressive motile sperm. Then, they must carefully review the Results section and clearly refer to one of these percentages instead of use the general term “motility”.
- Line 198. “kinetic traits”. What do the authors refer? Please, use an appropriate terminology in accordance with the analysis performed.
- Line 198. Please, refer to “sperm capacitation” instead of “capacitation of semen”.
- Line 201. “survival of semen”. Do the authors mean “sperm viability”? If so, please correct this term.
- Line 202. “increased”. Do the authors mean “increased percentage”? If so, please indicate.
- Figures 1A and 1E. Please, specify in the Y-axis the type of motility you measured, i.e. percentage of total or progressive motile spermatozoa.
- Line 209. “motility rate”. As previously indicated, this is not an appropriate term. Authors must refer to either the percentage of total or progressive motile spermatozoa.
- Lines 212-214. The content of these lines must be placed in a Table legend, instead of the Table title.
- Table 2. Please, reconsider if this table is necessary. I strongly recommend the authors to eliminate it and explain the information given in the text.
- Lines 216-219. The content of these lines must be placed in a Table legend, instead of the Table title.
- Lines 223-225. The content of these lines must be placed in a Table legend, instead of the Table title.
- Table 3. Authors provide the percentage of live cells and died cells. Please, note that most papers refer to sperm viability and they only provide the percentage of viable cells. Authors must eliminate the lines of the table referring to “died cells”.
Discussion section
- Lines 228-229. Please, revise this sentence and provide a clear message.
- Lines 234-238. Please, join these lines in a single paragraph.
- Lines 244-245. “…NGF may be determined by the ratio of p75NTR to TrKA”. What do the authors refer? In this manuscript authors did not measure this ratio, so this sentence is confusing. Please, revise it and provide a clear message.
- Lines 246-249. Please, revise the content of these lines and provide a clear message.
- Line 250. “…main semen traits” What do the authors refer? Please, note that authors did not analyse the semen quality, but the sperm quality. So, they must use an appropriate terminology.
- Lines 250-252. This review is unable to understand the meaning of this sentence. Please, review it and improve the message.
- Line 273. “…produce stably sperm with an early senescence”. What do the authors refer? Again, authors must review the sentence and provide a clear message.
Conclusions section
- Line 275. Please, specify why your results are so valuable for “fertility and reproduction issue”.
- Line 278. Please, refer to “some sperm quality parameters” instead of “distinct features”.
- Line 281 Please, refer to “in semen samples” instead of “in the semen”.
Reviewer 5 Report
This study focuses on a deeper understanding of the NGF and its interactions with the p75NTR on the male reproductive performance. By designing two separate experiments encompassing appropriate tecniques, the authors suggest that the NGF/receptor interaction could act as a predictor of semen quality - as such, my question is, what would be future directions in this research?
The study is scientifically sound and well written, perhaps a relatively low number of the animals involved in the experiments could be a limitation that should be discussed before coming to a definitive conclusion.
Round 2
Reviewer 1 Report
Dear authors,
Unfortunately, I do not find the work with the sufficient quality to be published in a journal as Cells. Since the journal bears an impact factor of 6.6 impact, both more important results and a more ambitious battery of evaluations are needed. In addition, the manuscript needs to be revised again by the authors as lacks many aspects that prevent the reader to figure out (1) why it is interesting to study NFK specifically in rabbit, (2) which evidences suggests that its function is affected by factors that may compromise male fertility and thus final litter production, (3) which effects have on NFK the current rabbit farming issues.... Many issues are unaddressed in the article. For this reason, I suggest the authors to modify and improve the article and to look for another journal where the manuscript fits better based on the impact of the results obtained and the interest of the species used to carry out the work.
Best regards,
PS. Please, find some more comments in the attached file.

Author Response
Reviewer 1 Responses
Unfortunately, I do not find the work with the sufficient quality to be published in a journal as Cells. Since the journal bears an impact factor of 6.6 impact, both more important results and a more ambitious battery of evaluations are needed. In addition, the manuscript needs to be revised again by the authors as lacks many aspects that prevent the reader to figure out (1) why it is interesting to study NFK specifically in rabbit, (2) which evidences suggests that its function is affected by factors that may compromise male fertility and thus final litter production, (3) which effects have on NFK the current rabbit farming issues.... Many issues are unaddressed in the article. For this reason, I suggest the authors to modify and improve the article and to look for another journal where the manuscript fits better based on the impact of the results obtained and the interest of the species used to carry out the work.
Best regards,
- Please, find some more comments in the attached file.
A: Considering the negative opinion expressed by the reviewer in both paper versions (original and revised), we have partially addressed the suggestions of the reviewer, giving priority to that of the other 4 reviewers if the comments were overlapping. This is the reason we have not considered all the comments in the first version.
See also general comment
- Do the authors have evidences of this??? Why do the authors imagine that?? Authors only noted that NGF varies between species but did not note evidencies suggesting other factors as possible affectors in NGF levels or function. Please, modify this paragraph because, in addition 60-64 was previoulsy described in 40-42.
A: We have worked on these aspects for a long time. In particular on the effect of rabbit age, collection rhythm and dietary plan on NGF receptors. A published review on rabbit has been added to support the sentence
- Which factors?? Why did the authors choose thes factors?? Previous bibiography identified some of them as modulators of NGF??
A: We added them
- How did authors induce in vitro capacitation and acorosome reaction??
Did authors verified that capacitation was properly achieved with a negative control??
A: As reported, we did not induce the capacitation, we analyzed the on time capacitive status of spermatozoa, within viable cells
- Please, change died with death cells.
A: The rev 4 and the language revision suggest to use the term "died cells". We prefer to use this term
- Capacitated cells?? Which pattern from the CTH assay was showed??Please, authors should be more specefic with this evaluation. In addition, did the authors evaluated phosphotyrosine of proteins as capacitation marker??
A: In the CTC assay method reported in M&M we analysed the Non-Capacitated cells when the fluorescence is evenly extended on the entire head; Capacitate when it is concentrated in the front of the head and the acrosome reaction when the fluorescence is distributed in a line at the level of the equator

Reviewer 4 Report
The authors introduced most of the changes requested in the previous revision, and so the quality of the manuscript has improved. Nevertheless, I still have some minor comments that authors must address before the acceptance of the manuscript for publication.
As a general comment I would like to note that the text has several typographical errors that must be corrected, and the English grammar also needs some improvement.
Other specific comments are listed below:
- Line 41. Please, correct “synthetized” by “expressed”.
- Figure 1. Authors provide in the figure the short title of Exp 1 but they do not of Exp 2. So, in order to improve the understanding of this Figure I recommend the authors to add a short title for Exp 2.
- Line 104. “each epididymal side”. I am not sure about what the authors mean. Do you refer to “each epididymal region”? If so, please correct this expression.
- Line 117. Please, refer to “viability” instead of “survival”.
- Line 136. Please, correct “acrosome reaction” by “acrosomal reaction”. Authors must revise the whole document to unify this term.
- Line 148. Please, correct “sperm cells sample” by “sperm cell samples”.
- Line 240. The sample size must be placed in the Table legend instead of the Table title.
- Line 254. Again, the sample size must be placed in the Table legend instead of the Table title.
- Figure 3. Please, specify the sample size in the Figure legend.
- Line 285. Please, correct “sperm cells type” by “sperm cell types”.
Author Response
Reviewer 4 Responses
We thank the reviewer for these additional suggestions that allow us to improve the quality of our manuscript.
- Figure 1. Authors provide in the figure the short title of Exp 1 but they do not of Exp 2. So, in order to improve the understanding of this Figure I recommend the authors to add a short title for Exp 2.
A: Following the reviewer's observation we emendated the Figure 1 with a short title of Exp 2“: Effect of exogenous NGF during storage”.
- Line 41. Please, correct “synthetized” by “expressed”.
- Line 104. “each epididymal side”. I am not sure about what the authors mean. Do you refer to “each epididymal region”? If so, please correct this expression.
- Line 117. Please, refer to “viability” instead of “survival”.
- Line 136. Please, correct “acrosome reaction” by “acrosomal reaction”. Authors must revise the whole document to unify this term.
- Line 148. Please, correct “sperm cells sample” by “sperm cell samples”.
- Line 285. Please, correct “sperm cells type” by “sperm cell types”.
A: We thank the reviewer for these observations. As requested, we correct all suggestions in the revised manuscript.
- Line 240. The sample size must be placed in the Table legend instead of the Table title.
- Line 254. Again, the sample size must be placed in the Table legend instead of the Table title.
- Figure 3. Please, specify the sample size in the Figure legend.
A: We thank the reviewer for these observations. As requested, we modified the table legend in the revised manuscript.
